# Identification and Characterisation of pST1023 A Mosaic, Multidrug-Resistant and Mobilisable IncR Plasmid

**DOI:** 10.3390/microorganisms10081592

**Published:** 2022-08-08

**Authors:** Carla Calia, Marta Oliva, Massimo Ferrara, Crescenzio Francesco Minervini, Maria Scrascia, Rosa Monno, Giuseppina Mulè, Cosimo Cumbo, Angelo Marzella, Carlo Pazzani

**Affiliations:** 1Department of Biology, University of Bari, Via Orabona, 4, 70125 Bari, Italy; 2Institute of Sciences of Food Production, National Research Council of Italy (ISPA-CNR), Via G. Amendola 122/O, 70126 Bari, Italy; 3Hematology and Stem Cell Transplantation Unit, Department of Emergency and Organ Transplantation (D.E.T.O.), University of Bari, 70124 Bari, Italy; 4Department of Basic Medical Sciences Neurosciences and Sense Organs Medical Faculty, University of Bari, Piazza G. Cesare Policlinico, 70124 Bari, Italy

**Keywords:** IncR plasmid, mosaic, mobilisable, antimicrobial resistance, IS*26*, pseudo-compound transposons, *sul3*-associated class 1 integron, *Salmonella* Southern European clone

## Abstract

We report the identification and characterisation of a mosaic, multidrug-resistant and mobilisable IncR plasmid (pST1023) detected in *Salmonella* ST1023, a monophasic variant 4,[5],12:i: strain of widespread pandemic lineage, reported as a Southern European clone. pST1023 contains exogenous DNA regions, principally gained from pSLT-derivatives and IncI1 plasmids. Acquisition from IncI1 included *oriT* and *nikAB* and these conferred the ability to be mobilisable in the presence of a helper plasmid, as we demonstrated with the conjugative plasmids pST1007-1D (IncFII) or pVC1035 (IncC). A *sul3*-associated class 1 integron, conferring resistance to aminoglycosides, chloramphenicol and trimethoprim-sulphonamides, was also embedded in the acquired IncI1 DNA segment. pST1023 also harboured an additional site-specific recombination system (*rfsF*/*rsdB*) and IS elements of the IS*1*, IS*5* (IS*903* group) and IS*6* families. Four of the six IS*26* elements present constituted two pseudo-compound-transposons, named PCT-*sil* and PCT-Tn*10* (identified here for the first time). The study further highlighted the mosaic genetic architecture and the clinical importance of IncR plasmids. Moreover, it provides the first experimental data on the ability of IncR plasmids to be mobilised and their potential role in the horizontal spread of antimicrobial-resistant genes.

## 1. Introduction

Plasmids are self-replicating, extra-chromosome genetic elements and are considered an important driving force of bacteria evolution as they contribute towards generating genetic variability and also simply provide selective advantages, such as antimicrobial resistance [1]. The latter is of great importance since resistance to most available antimicrobial classes has been recognised as an emerging health problem on a worldwide scale. Indeed, the global increase in multidrug-resistant (MDR) bacteria has drastically been reducing the range of antimicrobials available to treat bacterial infections. This has made antimicrobial resistance to bacteria the major cause of death worldwide, as recently reported by a comprehensive survey on this, covering over 204 countries and territories and published in The Lancet journal [2].

Antimicrobial resistance genes can be embedded within genetic elements, such as transposons, compound transposons and integrons (mainly of class 1), that are often carried by plasmids (particularly in *Entebacterales*), which, in turn, greatly contribute to antimicrobial resistance spreading and the insurgence of multidrug-resistant bacteria [3,4]. Indeed, many bacteria genomes contain multiple plasmids whose persistence is achieved by vertical transmission to daughter cells and (if conjugative) by transmission through cell-to-cell conjugation. The ability to be horizontally transferable represents an evolutive advantage in that it would further extend the host range, thus, increasing the general level of long-term persistence in bacteria population [5,6].

Plasmids can generally be classified into different types according to their replication (replicon type) or mobility (MOB typing) loci [7,8]. Plasmids sharing the same replication system are unable to co-exist stably within the same host cell and are clustered in the same replicon type (group of incompatibility or Inc group). MOB typing, based on relaxase protein phylogenies, allows classification of transmissible plasmids into MOB families. IncR is a relatively recent replicon type, first reported in 2009 and identified in pK245, a plasmid harboured by a multidrug-resistant *Klebsiella pneumoniae* strain [9]. IncR plasmids are not included in the MOB typing system as they do not contain a relaxase gene. Additionally, they do not possess conjugational transfer genes and, thus, are not conjugative. Since their first identification, IncR plasmids have been isolated the world over, mainly from clinical multidrug-resistant strains [3]. The IncR replication and maintenance systems are principally composed of *repB* (replication initiation) and its iterons, *parAB* (partition) and *vagCD* (toxin–antitoxin). In addition to their core backbone, IncR plasmids may carry various accessory modules, often conferring resistance to different classes of antimicrobials that extend the size of these up to 160 Kb [10].

In this study, we report the identification and characterisation of a mosaic, multidrug-resistant and mobilisable IncR plasmid (pST1023) detected in the *Salmonella enterica* subsp. *enterica* serovar 4,[5],12:i: strain ST1023. ST1023 belongs to the widespread pandemic lineage reported as a Southern European clone [11] and the finding of a mobilisable IncR plasmid, experimentally demonstrated here for the first time, is of concern for its potential role in the spread of antimicrobial-resistance genes.

## 2. Materials and Methods

### 2.1. Bacteria Isolates, Antimicrobial Susceptibility Testing and Mobilisation Experiment

The STMV ST1023 is a clinical strain isolated in Southern Italy in 2008 [12]. Antimicrobial susceptibility tests were performed as reported previously [13]. The antimicrobials were: ampicillin (Ap), chloramphenicol (Cm), streptomycin (Sm), sulphamethoxazole (Su), tetracycline (Tc) and trimethoprim (Tp). MIC (minimal inhibitory concentration) to silver nitrate was determined by the broth microdilution method with Mueller–Hinton (MH) broth according to the Clinical and Laboratory Standards Institute (CLSI) guidelines. AgNO_3_ concentrations ranging between 2 and 512 μg/mL were tested [14]. ST1023 silver-resistant mutants were selected by plating saturated cultures (≥10^9^ cfu/mL) onto MH agar containing AgNO_3_ up to 256 µg/mL.

Conjugation experiments were performed at 37 °C as described previously [15]. Antimicrobial concentrations were: Ap 100 µg/mL, Cm 25 µg/mL, nalidixic acid (Nx) 50 µg/mL, rifampicin (Rf) 100 µg/mL, Sm 100 µg/mL, Su 600 µg/mL, Tc 20 µg/mL and Tp 30 µg/mL. CSH26 Nx or DH5α Rf strains were used as recipients. The frequency of transfer (mean number of transconjugants per donor) was determined in three or more independent experiments and the standard deviation (SD) calculated.

### 2.2. DNA Sequencing, Assembly and Annotation

Total genomic DNA was extracted by the cetyl trimethylammonium bromide method [16]. Plasmid DNA was isolated as described previously [17]. About 1 μg of DNA was fragmented by using the Ion Shear™ Plus Reagents Kit (Life Technologies, a part of Thermo Fisher Scientific Inc., Waltham, MA, USA), followed by barcoded adapter ligation using the Ion Xpress™ Barcode Adapters (Life Technologies, a part of Thermo Fisher Scientific Inc.) and Ion Plus Fragment Library Kit (Life Technologies, a part of Thermo Fisher Scientific Inc.) according to the manufacturer’s protocol. The library size was selected (~400 bp) using E-Gel^®^ SizeSelect™ 2% Agarose Gel (Invitrogen, Carlsbad, CA, USA). Library concentrations were quantified using the Qubit dsDNA HS Assay Kit (Life Technologies, Waltham, MA, USA). Sequencing template was prepared by using the Ion 520 & 530 Kit-OT2 kit (Thermo Fisher Scientific Inc.) and then sequenced on an Ion 520 Chip using an Ion GeneStudio S5 System (Thermo Fisher Scientific Inc.). Raw data were quality filtered and assembled by using the SPAdes assembler version 3.15.4 [18] and the included pipeline plasmidSPAdes (--plasmid). pST1023 plasmid DNA was sequenced using MinION (Oxford Nanopore Technologies, Oxford, United Kingdom). Sequencing library was prepared using Rapid Sequencing Kit SQK-RAD004 (Life Technologies, a part of Thermo Fisher Scientific Inc) and 500ng of DNA following the manufacturer’s instructions. The sequencing was performed using the R9.4.1 flongle flowcell FLO-FLG001 for 24h, according to the information provided by the manufacturer (https://store.nanoporetech.com/eu/flongle-flow-cell-pack.html, last access on 18 February 2022). The basecalling of the raw signals from the sequencing run was performed with Guppy v.5.0.11 (Oxford Nanopore Technologies) by the r9.4.1_450bps_hac model. Only the fastq files in the Guppy directory “pass”, considered as high-quality reads, were used for the assembly. De novo genome assembly of basecalled reads was performed using Canu v.2.2 with default parameters [19]. The complete genome sequence was deposited in NCBI and annotated by the NCBI Prokaryotic Genome Annotation Pipeline (https://www.ncbi.nlm.nih.gov/genome/annotation_prok/, last access on 1 July 2022 [20]. The genome of ST1023 is publicly available under the Bioproject ID PRJNA854888 in GenBank. Plasmid sequences of pSLT (GenBank Acc. N° AE006471.2)) and pST1030-1A (GenBank Acc. N° MT507877) were used for comparison.

### 2.3. Bioinformatic Analysis

Multilocus sequence type (MLST) of ST1023 was ST19. It was assigned using the PubMLST scheme for *Salmonella* spp. (https://pubmlst.org/; November 2021) with the following results: aroC10, dnaN7, hemD12, hisD9 and purE5 [21]. Similarity searches were performed using the BLASTN algorithm of the NCBI Web BLAST (https://blast.ncbi.nlm.nih.gov/Blast.cgi; March 2022) and the entire or selected regions of pST1023 sequence as query. Results were graphically depicted by SnapGene (http://www.snapgene.com/; November 2021) and Adobe Illustrator (https://www.adobe.com/it/; November 2021) Plasmid replicon type was determined using the PlasmidFinder v.2.0.1 (https://cge.cbs.dtu.dk/services/PlasmidFinder/; November 2021) [22,23]. ISFinder was used to identify complete or partial insertion sequences and mobile genetic elements (http://www-is.biotoul.fr; March 2022) [24]. Tandem repeats, period size and copy number of iterons were detected by Tandem repeats finder (https://tandem.bu.edu/trf/trf.html; March 2022) [25].

## 3. Results

### 3.1. Genome Sequence of ST1023 and Context of Resistance Genes

ST1023 is part of a collection of *Salmonella* MDR clinical strains isolated in Southern Italy from 2006 to 2012 [12]. The genome of ST1023 was sequenced and analysis of the *aroC*, *dnaN*, *hemD*, *hisD*, *purE*, *sucA* and *thrA* housekeeping gene classified ST1023 in the MLST group ST19. Based on the absence of the *fljB* gene, detected by PCR, ST1023 was assigned to the monophasic variant 4,[5],12:i:- [26]. The genome sequence of *Salmonella* Typhimurium LT2 (GenBank Acc. N° AE006468.2) was used as reference for comparison with that of ST1023. A large chromosomal deletion of 70,456 bp was detected in ST1023: it started at 435 bp downstream of STM2692 and ended in the inverted repeat sequence *hixR* of the segment H. The segment H, required for phase variation, contains the promoter for *fljB* and the *hin* gene, encoding a DNA invertase necessary for inversion of the H segment. The chromosomal deletion included the prophage Fels-2 genome, the region from STM2741 to STM2769 and the *fljAB* operon, whose absence accounted for the monophasic variant 4,[5],12:i:-. In the place of the 70,456 bp deletion, there was a fragment of 5349 bp. It was composed of six Open Reading Frames (ORFs), named ORF1 to ORF6. ORF2 showed partial homologies with both STM1053 and STM1054 (genes of the prophage Gifsy-2). ORF3 (549 bp) was partially homologous with STM1997, a gene reported to encode for a component (UmuC) of the DNA polymerase V. ORF4 was homologous to STM2704. ORF5 and ORF6 showed partial homologies with STM2705 and STM2706, respectively. STM2704, STM2705 and STM2706 are part of the Fels-2 prophage genome. ORF1 had no homology with any LT2 genes.

ST1023 was resistant to Cm, Sm, Su, Tc and Tp encoded by *cmlA1*, (*aadA1*, *aadA2*), *sul3*, *tetA*(*B*) and *dfrA12*, respectively. *aadA1*, *aadA2*, *cmlA1*, *dfrA12* and *sul3* were part of a sul3-associate class1 integron, while *tetA*(*B*) was part of a pseudo-compound transposon (see below). The antimicrobial-resistance genes were harboured by an IncR plasmid, named pST1023. The genomic and phenotypic features detected for ST1023 were consistent with those characterising the pandemic serotype 4,[5],12:i:- lineage referred to as the Southern European clone, a monophasic variant of *S*. Typhimurium that has emerged as a major global cause of non-typhoidal disease in animals and humans [11,27,28]. pST1023 also harboured a second pseudo-compound transposon harbouring a *sil* operon (see below). However, ST1023 was found sensitive to AgNO_3_ (MIC ranging from 4 to 8 µg/mL)

### 3.2. pST1023 Genetic Organisation

pST1023 consisted of 120,313 bp with an average G+C content of 51.4% (Figure 1). The assembled sequence was confirmed by comparing the *BamHI*, *ClaI*, *HindIII* and *KpnI* restriction profiles generated in silico with those obtained from restrictions of plasmid DNA. Based on computational analysis for functional gene prediction, three major regions designated IncR backbone, I1 and SLT (the last two traceable to plasmids IncI and pSLT derivatives, respectively) were identified. pST1023 also harboured a Tn*21*-derived element lacking in the *mer* operon (termed Tn*21*-Δ*mer*), two IS*26*-bound pseudo-compound transposons [29] carrying a Tn*10*-derived and a *sil* gene cluster, named PCT-Tn*10* and PCT-*sil*, respectively, and IS elements (or their isoforms) of the families IS*1*, IS*5* and IS*6* (Appendix A).

The IncR backbone included *repB* (that encodes the replication initiation protein RepB) and its iterons (composed of 36 bp present in 10,4 copies), *parAB* (a partition system encoding the ATPase protein ParA and the centromere-binding protein ParB), *umuCD* (that encodes an error-prone DNA polymerase V, a key contributor in the SOS response), *retA* (a group IIB intron-encoding reverse transcriptase), *rfsF*-*resD* (site-specific recombination system) and *vagCD* (encoding a type II toxin–antitoxin (TA) system, consisting of VagC antitoxin and VagD toxin) [10,30]. *vagCD* was inversely oriented and separated from *rfsF* by region I1 (57,754 bp). Region I1 was composed of a remnant of fragment C (7.1 Kb, see below), a Tn*21*-Δ*mer* (14.6 Kb) and a 32.5 Kb fragment (called I1-*oriT*); the last was linked to sequences from the leading region and part of the conjugative region of IncI1 plasmids [31]. The fragment C has mainly been identified in IncI1 plasmids, some of which had a Tn*21*-derived transposon (like the Tn*21*-Δ*mer* but retaining the *mer* operon) inserted into *ydfA* [17]. The Tn*21*-Δ*mer* also mapped next to 5′-Δ*ydfA* and was composed of *tnpA* (transposase), *tnpR* (resolvase), *tnpM* (interrupted by a *sul3*-associated class 1 integron, carrying the array *dfrA12*-*orfF*-*aadA2*-*cmlA1*-*aadA1*-*qacH*) and *mef* (interrupted by the IS*26*_1_). The fragment I1-*oriT* was composed of: I) the *impCAB* operon (*impB* was interrupted by the IS*26*_1_) encoding (based on the homology to *umuCD*) an error-prone DNA repair system [32]; II) the *psiAB* operon (inhibition system of the SOS response that during conjugation prevents LexA autocleavage catalysed by RecA) [33]; III) the *ardA* gene, encoding an anti-restriction and anti-modification protein that prevents cleavage at foreign DNA entering a new bacteria host [34]; IV) the relaxation complex, composed of the *oriT* sequence (origin of transfer) and the *nikAB* gene cluster (encoding the *oriT*-specific DNA binding protein NikA and the relaxase NikB, respectively) [31]; V) the *trbABC* operon essential for conjugational transfer of IncI1 plasmids (TrbA and TrbC are key elements in delivery DNA molecules to be secreted across the T4SS) [35]; VI) the toxin/antitoxin system *pndABC* (Hok/Sok TA family), composed of a stable mRNA encoding a toxin (e.g., PndA), a more unstable antisense mRNA (e.g., PndB encoded mRNA) and *pndC,* which modulates *pndA* expression by promoting its translation [35]; VII) the entry exclusion system composed of *excA* and *traY* [36]; VIII) the *traX* gene encoding an inner membrane conjugal transfer pilus acetylation protein; and IX) the traW gene (encoding a lipoprotein) interrupted by the IS*26*_2_-v1 (classified by the recent proposed nomenclature as IS*26*-v1 in that differing from IS*26* sequence for three nucleotides, of which two caused the single amino acid substitution G184N in the catalytic domain of Tnp26 [37]). Between *pndABC* and *excA*, there was an IS*1N* element flanked by an 8 bp (CGATAGCT) target site duplication (TSD). Between region I1 and *vagCD,* we mapped a multiple ΔIS locus (locus A) composed of a ΔIS*102* element (IS*5* family), truncated by an IS*2* element (IS*3* family), truncated by an IS*Ec15* (IS*3* family), truncated by a Tn*5393* (Tn*3* family), interrupted by the IS*26*_2_-v1 (Appendix A). Downstream of *vagCD* was the region SLT that included an *rfsF*-RsdB system, the type II TA system *ccdAB* [38,39] and the *spvABCD* operon (associated with strains that cause non-typhoid bacteraemia), with its positive regulator *spvR* [40]. A PCT-Tn*10* (flanked by IS*26*) was inserted into *spvC,* splitting the SLT region into two fragments, of which one carried the 3′-Δ*spvC* and one, inversely oriented, the 5′-Δ*spvC*. TSD of 8 bp (CTTTAAAG) was detected downstream of both 3′-Δ*spvC* and 5′-Δ*spvC*. The PCT-Tn*10* harboured three genes (*jemA*, *jem* B and *jemC*) unrelated to tetracycline resistance, genes encoding tetracycline resistance (*tetA*, *tetC* and *tetD*) and *tetR* encoding tetracycline transcriptional repressor [41]. *tetD* was interrupted by the IS*26*_4_-v1 that caused loss of IS*10*-R of the ancestral Tn*10*; of the IS*10*-L, only 66 bp were retained. Between the IS*26*_3_ and ΔIS*10*-L being mapped were a locus named locus B, composed of a ΔTnEc*1*, a ΔIS*1*, a fragment of 68 bp and an IS*903B* element (Appendix A). Following the SLT region mapped PCT-*sil* (flanked by IS*26*_4_-v1 and IS*26*_5_), composed of the *silE*, *silC*, *silF*, *silB*, *silA*, *silG* and *silP* structural genes and the two-component silver-responsive transcriptional genes *silR* and *silS* [42]. The mechanism of silver resistance includes a cation sequestration in the periplasm (via SilE and SilF), an active silver efflux (via SilCBA efflux transporter and a putative P-type ATPase transporter SilP) and a signal transduction system, mediated by the sensor histidine kinase SilS and the response regulator protein SilR. The *sil* genes cluster was flanked by two ΔIS*1* (referable in the IS finder database as either IS*1A*, IS*1R* or IS*1S*) interrupted by the IS*26*_4_-v1 and IS*26*_5_. Downstream of (554 bp) *silP,* there was an IS element (IS*Kpn74*) of the IS*5* family subgroup IS*903* [43]. 

Between PCT-*sil* and the IncR backbone, there was a region of about 7227 bp where the 4651 bp just upstream of *retA* were found linked to many IncR plasmids (see below). This last region (termed N) was characterised by the two IS elements IS*903B* (subgroup IS*903*, family IS*5* [44,45]) and IS*1X3* (family IS*1* [46], and a sequence of the IncN backbone composed of a remnant of *repA* (encoding the IncN replication initiation protein) and its iterons (composed of 37 bp present in 24.3 copies representing the RepA-binding site) [47,48]. The 2576 bp, spanning from PCT-*sil* to the N region, included an IS*1X2* (family IS*1*) and an open reading frame of 741 bp, encoding for a putative tyrosine recombinase. Tyrosine recombinases constitute a large family of proteins involved in different biological processes (e.g., post-replicative segregation of plasmids) [49].

### 3.3. pST1023 A Mosaic Plasmid

Plasmids can be shaped by homologous recombination, non-homologous end joining and rearrangements of unknown mechanisms that lead to the generation of mosaic plasmids [1]. We then explored whether pST1023 might fall within this class of plasmids by searching the singular regions or their partial sequences (termed sub-fragments) for high similarity (100% coverage and ≥99.9% identity) (Figure 2).

The overall structure of region I1 was not detected in any plasmid. However, when the search was carried out for its sub-fragments C and I1-*oriT*, or the Tn*21*-Δ*mer*, a number of hits was found. Fragment C was identified in IncI1-complex (32), IncFII (4) and IncFIIs-FIB (1) plasmids and in plasmids of an unknown incompatibility group (2). The Tn*21*-Δ*mer* was found in the Tn*21*-derived element (that included sequences identical to Tn*21*-Δ*mer*), harboured by plasmids IncX1 (13), IncI1 (4), IncH1 (1), IncR (1), IncFII/FIB (1), IncFIIs/IB (1) and ColE1 (1). The fragment I1-*oriT* (excluding the IS*1N* sequence) was detected in 3 IncI1 plasmids (pST1030-1A and its derivatives pST1030-1B and pST1030-1C, GenBank Acc. N° MT507877, MT507879 and MT507880, respectively). Only the last three IncI1 plasmids harboured the sub-fragments C and I1-*oriT* and the Tn*21*-Δ*mer*, albeit differently organised along their sequences. The SLT fragments flanking PCT-Tn*10* were detected in many pSLT-derived plasmids. Overall, these data are consistent with a classification of pST1023 as a mosaic plasmid. 

We also tried to track down IncR plasmids that shared markers with pST1023, whose common genetic organisation might mirror possible common evolutionary steps. Three DNA sequences were selected: that of region N (already reported in many IncR plasmids) and those of loci A and B (whose genetic structures were the output of precise events of insertions and/or recombination of IS and Tn elements) (Appendix A, respectively). Seventy-four IncR sequences were found to harbour high-similar regions N (100% coverage and ≥99.9% identity) linked to the IncR backbone. Of the 74 retrieved sequences, 18 lacked *vagCD* (DNA homology ended to *rfsF*), 2 had *vagCD* separated from *rfsF* by DNA fragments different in size and genetic information and 48 retained *vagCD*, next to *rfsF*, linked to sequences (ΔIS*102*-ΔI*S2* and *ISEc15* or Δ*ISEc15*) related to locus A as follows: 1, pF18S020 (GenBank Acc. N° CP082454), held an intact IS*Ec15*, 41 had IS*Ec15* disrupted (same nucleotide position) by an IS*Kpn60* and 6 the IS*Ec15* disrupted (same nucleotide position as in pST1023, see below) by a Tn*5393* (GenBank Acc. N° CP057379, CP058068, CP064111, KM877517, LR890355 and LR890753). In pST01023, the IS*Ec15* (disrupted by Tn*5393*) was then interrupted by an IS*26*_2_-v1 element, thus, generating the structure of locus A. In pST1023, the region, including *vagCD* (3320 bp), was inversely oriented and separated from *rfsF* by the region I1.

In addition to locus A, the two plasmids with GenBank Acc. N° CP057379 and CP058068 (plasmids pRHB28-C14_2 and pRHB02-C19_6, respectively) also harboured an identical locus B, localised next to the remnant of gene *brxL* (nucleotide position 7467-8555 in GenBank Acc. N° CP058068). Locus B might have originated in a pKqq_18A069_2 (GenBank Acc. N° CP084820) -like plasmid, where a ΔIS*1*-Tn*Ec1* was separated from the *estP* (nucleotide position 24.426-27.002) gene by a 55 bp fragment. Insertion of IS*903B* into *estP* (13 bp from its 3′ end) generated the multi-IS locus B (IS*903B* separated from ΔIS*1*-ΔTnEc*1* by 68 bp that included the 13 bp 3′*estP*) found in the plasmid p1506-1 (GenBank Acc. N° CP059289). However, locus B in pRHB28-C14_2 and pRHB02-C19_6 was found next to the *brxL* gene or, as in pST1023, within the PCT-Tn*10*. The origin of this different genetic localisation may result from recombination events, possibly mediated by IS*903B* [50]. Plasmids pRHB28-C14_2 and pRHB02-C19_6 were virtually identical to each other (99.8% identity, coverage 100%), had a size (about 43 Kb) smaller than that of pST1023 and, interestingly, harboured a complete Tn*10*. These data together suggest that pRHB28-C14_2, pRHB02-C19_6 and pST1023 might have shared possible common evolutionary steps.

### 3.4. pST1023 Mobilisation

Conjugation experiments to assess the self-horizontal transfer of pST1023 failed to detect any transconjugants (detection frequency less than 1 × 10^−9^) (Table 1). Detection of the IncI1 relaxation complex (*oriT* and *nikAB*) in pST1023 prompted us to investigate its possible mobilisation, mediated by the copresence of conjugative plasmids. The self-transmissible plasmids pST1007-1D (IncFII, encoding resistance to Ap-Sm-Su ([17]) and pVC1035 (IncC, encoding resistance to Ap-Kn-Sm-Su-Tc ([51]) were introduced (by conjugation) into ST1023, generating the transconjugants BA3A and BA3B, respectively. Results from mating types using BA3A or BA3B as donor and CSH26 Nx as recipient highlighted the mobilisation of pST1023, mediated by either pST1007-1D (transconjugant BA3D) or pVC1035 (transconjugant BA3F), with a mean frequency of 8 × 10^−5^ and 1.6 × 10^−4^, respectively. The presence of pST1023 in BA3D and BA3F was also assessed by PCR and enzyme restrictions of plasmid content.

## 4. Discussion

Plasmids are genetic elements that may supply a valuable and variable gene pool. Moreover, features, such as maintenance over generations into daughter cells and ability to transfer from hosting to recipient cells, make plasmids a driving force of bacteria ecology and evolution [1]. Plasmids are subject to molecular evolution through genetic reassortments, mainly occurring among plasmids themselves or with other genetic elements, such as integrons, transposons, etc. In this respect, a set of plasmids, termed “mosaic”, has recently been the topic of growing scientific interest, as shown by the number of studies published on this topic [52,53,54]. Identification and characterisation of mosaic plasmids can, indeed, help to better assess the extent of molecular dynamics (both known and yet to be discovered) on plasmid evolution and the role played by this set of plasmids in the spread of genes conferring selective advantages, including those encoding antimicrobial resistance [55,56].

pST1023 is a mosaic, multidrug-resistant and mobilisable IncR plasmid, harboured by the clinical strain ST1023, a monophasic variant 4,[5],12:i:- of the widespread pandemic lineage, reported as a Southern European clone. pST1023 acquired exogenous DNA fragments from other plasmids, mostly IncI1 and pSLT-derivatives. Some of these fragments conferred resistance to different classes of antimicrobials. One, probably acquired from an IncI1 plasmid, carried a *sul3*-associated class 1 integron (embedded into a Tn*21*-Δ*mer*) that conferred resistance to aminoglycosides, chloramphenicol and trimethoprim-sulphonamides. The second, PCT-Tn*10*, conferred resistance to tetracycline. Other acquired fragments, also from IncI1 plasmids, carried *oriT* and *nikAB* that allowed pST1023 to be horizontally transferred by the copresence of a conjugative plasmid, as we demonstrated with the IncFII and IncC plasmids pST1007-1D and pVC1035, respectively.

Acquisition of some of pST1023′s exogenous DNA fragments could be the result of transposase activity of different IS elements, such as IS*26* and IS*903*B [57]. The presence of six IS*26* elements, of which four were organised into two pseudo-compound transposons (PCT-Tn*10* and PCT-*sil*), might be consistent with that possibility and also reinforce the role played by this class of IS in shaping plasmids and spreading ARGs. In this respect, it is worth noting that two IS*26s* (IS*26*-2 and IS*26*-4) were classified as IS*26*-v1, an IS*26* variant with enhanced activity [37], and that IS*26-*4 was part of PCT-Tn*10*. To the best of our knowledge, this is the first report on the detection of this pseudo-compound transposon and this finding poses concern regarding the diffusion of tetracycline resistance.

Silver ions and silver-based compounds (e.g., nanoparticles) are well-known antimicrobial agents used in clinical and medical practice, as well as in agricultural and industrial products [58,59]. As a result, isolation of silver-resistant bacteria has recently been increasing, making silver resistance an emerging problem [60,61]. Moreover, cryptic silver resistance that can be readily activated by single missense mutation in *silS* (encoding the sensor histidine kinase) has widely been reported for some genera (e.g., Enterobacter and Klebsiella) [62,63]. ST1023 is sensitive to AgNO_3_; however, spontaneous ST1023 silver-resistant mutants were selected on MHA plates added with AgNO_3_ up to 256 µg/mL. Analysis of these mutants is in progress to assess the possible cryptic silver resistance of ST1023. Anyway, the finding of *sil* operon organised into an IS*26*-bound transposon (PCT-*sil*) is undoubtedly of concern, since IS*26* elements are encountered with increasing frequency in genomes (especially plasmids) of clinically relevant bacteria [64].

Mechanisms of recombination, other than those mediated by transposases, for instance, site-specific recombination systems (*rfsF*-RsdB and *rfsF*-ResD), may also have contributed to the acquisition of the exogenous DNA fragments SLT and I1, respectively. In this regard, it is worth mentioning the finding of some IncR plasmids with *vagCD* separated from the core backbone through the insertion of different DNA fragments mapping just downstream *rfsF*. In pST1023, the *rfsF*-ResD system might have been involved in acquisition of the region I1 from IncI1 plasmids, such as pST1030-1A. Acquisition from IncI1 plasmids (like pST1030-1A) and involving the *rfsF*-ResD system has also been reported for the mosaic IncFII-plasmid pST1007-1A. It is noteworthy that pST1007-1A, pST1023 and pST1030-1A were all isolated from *Salmonella* clinical cases, which occurred in Apulia in the three-year period 2006–2008.

The broad host range of IncR plasmids has been linked to the possibility that these plasmids are mobilisable [3,65]. However, sequencing results showed that IncR plasmids did not possess relaxase genes nor origin of transfer sequences (*oriT*) and the mobilisation of IncR plasmids remains to be proven. In this study, we report the first experimental data on the mobilisation of a mosaic, multidrug-resistant IncR plasmid, pST1023, containing the IncI1 *oriT*-*nikAB* region. The study also attempts to outline possible evolutionary steps shared with other IncR plasmids.

## Figures and Tables

**Figure 1 microorganisms-10-01592-f001:**
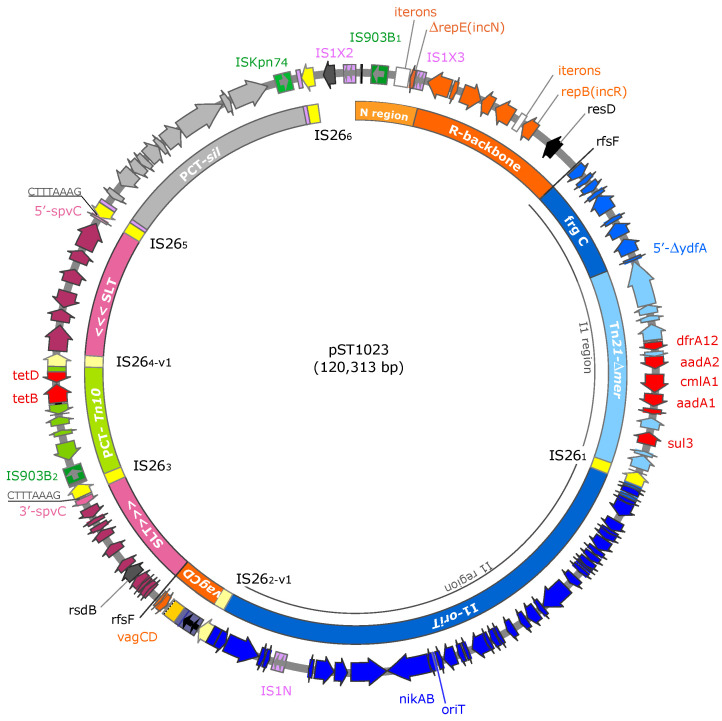
Physical map of pST1023. Outer ring: selected annotated open reading frames, IS elements and specific loci are shown. Arrows indicate the 5′ to 3′ transcription direction. Inner ring: the different regions that constitute the mosaic pST1023 are shown with coloured boxes: IncR backbone (red orange), Fragment C and I1-*oriT* (blue), SLT (dark terracotta), Tn*21*-Δ*mer* (light blue), PCT-Tn*10* (apple green) and PCT-*sil* (grey). IS*26* (yellow) and IS*903B* (green) are numbered.

**Figure 2 microorganisms-10-01592-f002:**
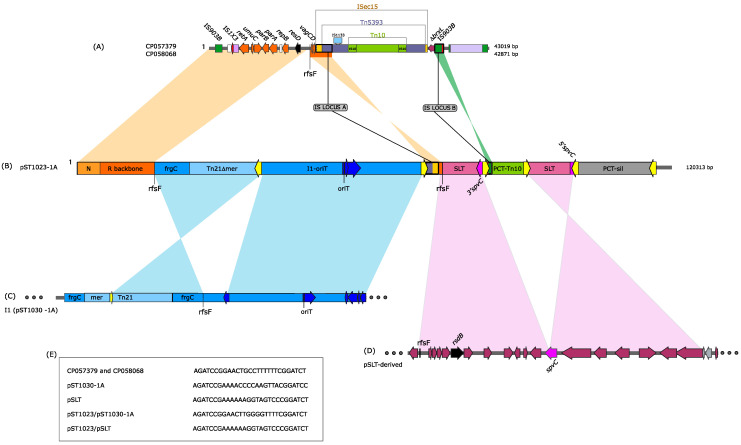
The mosaic plasmid pST1023. Shaded regions, with nucleotide identity ≥99%, are marked by colours. (**A**) IncR plasmids sharing the N region, R backbone, loci A and B. (**B**) Linear map of pST1023. The different regions that constitute the mosaic pST1023 are shown with coloured boxes. IS*26* elements are in yellow. (**C**) Regions of IncI1 plasmids (such as pST1030-1A) shared with pST1023. (**D**) Regions of pSLT-derived plasmids shared with pST1023. (**E**) Sequences of *rfsF* sites are reported.

**Table 1 microorganisms-10-01592-t001:** Conjugation experiments and mobilisation of pST1023.

Strain	Resistance(s) ^a^	Resistance Genes	Plasmid	Transconjugant (Plasmid)	Resistance Genes Transferred by Conjugation	Frequency of Conjugation (SD) ^d^
ST1023	CmSmSuTcTp	*dfrA12-aadA2-cmlA1-aadA1-sul3-tetB-tetC*	pST1023	ND	none	none
BA3A	CmSmSuTcTp	*dfrA12-aadA2-cmlA1-aadA1-sul3-tetB-tetC*	pST1023	BA3C		
	ApSmSu	*bla* _TEM_ *-strAB-sul2*	pST1007-1D ^(b)^	(pST1007-1D)	*blaTEM-strAB-sul2*	3.0 (±3.9) × 10^−1^

				BA3D		
				(pST1007-1D)	*bla_TEM_-strAB-sul2; tetB*	8.0 (±0.2) × 10^−5^
				(pST1023)	*dfrA12-aadA2-cmlA1-aadA1-sul3-tetB-tetC*	
BA3B	CmSmSuTcTp	*dfrA12-aadA2-cmlA1-aadA1-sul3-tetB-tetC*	pST1023	BA3E	*blaTEM-aphaI-strAB-sul2-tetD*	1.6 (±0.0) × 10^−1^
	ApKnSmSuTc	*bla* _TEM_ *-aphaI-strAB-sul2-tetD*	pVC1035 ^(c)^	(pVC1035)		

				BA3F		
				(pVC1035)	*blaTEM-aphaI-strAB-sul2-tetD*	1.6 (±0.0) × 10^−4^
				(pST1023)	*dfrA12-aadA2-cmlA1-aadA1-sul3-tetB-tetC*	

^a^ Ampicillin (Ap); Kanamycin (Kn); Chloramphenicol (Cm); Streptomycin (Sm); Sulfamethoxazole (Su); Tetracycline (Tc); Trimethoprim (Tp). ^b^ Inc I1 plasmid. ^c^ IncC plasmid. ^d^ Values represent the mean of transconjugants per donor (SD stands for Standard Deviation). ND: not detectable.

## Data Availability

The genome of ST1023 is publicly available under the Bioproject ID PRJNA854888 in GenBank.

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
