# Peer review of "Identification and Characterisation of pST1023 A Mosaic, Multidrug-Resistant and Mobilisable IncR Plasmid"

_microorganisms, 2022, doi:10.3390/microorganisms10081592_

Round 1
Reviewer 1 Report
This paper reports the identification and characterisation of IncR plasmid from Salmonella ST1023. An interesting finding is the mobilisation of a mosaic, multidrug resistant 489 IncR plasmid, pST1023, containing the IncI1 oriT-nikAB region. In this paper, Pct-tn10 compound transposon was identified for the first time. The results may be helpful for understanding the multidrug resistance of Salmonella. The reviewer think this paper can be accepted after addressing the following questions.
1. It is reported that the IncR plasmid is co-integrated with the incx3 plasmid mediated by IS26 to obtain a binding element, which transmits the drug resistance gene in CRKP. see Journal of Antimicrobial Chemotherapy, 2021, 76, 2017–2023. this reference should be cited.
2. Isn't the number of base pairs an integer? How can there be decimals? Please check the number of bp.
Author Response
The paper has been modified in observance with the comments. We are sincerely grateful to the reviewer which helped us to significantly improve our manuscript. The changes have been indicated using the “Track Changes” function tool in Word.
Reviewer # 1: This paper reports the identification and characterisation of IncR plasmid from Salmonella ST1023. An interesting finding is the mobilisation of a mosaic, multidrug resistant 489 IncR plasmid, pST1023, containing the IncI1 oriT-nikAB region. In this paper, Pct-tn10 compound transposon was identified for the first time. The results may be helpful for understanding the multidrug resistance of Salmonella. The reviewer think this paper can be accepted after addressing the following questions.
- It is reported that the IncR plasmid is co-integrated with the incx3 plasmid mediated by IS26 to obtain a binding element, which transmits the drug resistance gene in CRKP. see Journal of Antimicrobial Chemotherapy, 2021, 76, 2017–2023. This reference should be cited.
Reply 1: this reference has been added
- Isn't the number of base pairs an integer? How can there be decimals? Please check the number of bp.
Reply 2: corrected
Reviewer 2 Report
The Authors present the identification and characterization of a IncR plasmid (pST1023) associated with multi-drug resistance. However, the Introduction does little to introduce us to this problem, namely the problem of microbial resistance. Expand the Introduction with this information, then it will be juicier.
The Authors should make the necessary corrections to the manuscript: the name Enterobacteriaceae is no longer used, only the family is now called Entebacterales.
Salmonella should be in italics.
In writing the name: Salmonella enterica subsp. enterica does not need to use the full word "subspecies".
When using the name Salmonella Typhimurium, the full name should be used for the first time, ie Salmonella enterica subsp. enterica serovar Typhimurium. The short name may be used only in the next use. Please, let the Authors refer to the correct naming of the bacteria.
The signature to Fig 1 is on a different page than the Figure itself.
No caption for Table 1. The name "Table 1" is in the Result description string.
Table 1 is broken, the column widths should be changed.
With these minor corrections, the manuscript will be more accessible to the reader.
Best regards
Author Response
The paper has been modified in observance with the comments. We are sincerely grateful to the reviewer which helped us to significantly improve our manuscript. The changes have been indicated using the “Track Changes” function tool in Word.
Reviewer # 2: The Authors present the identification and characterization of a IncR plasmid (pST1023) associated with multi-drug resistance. However, the Introduction does little to introduce us to this problem, namely the problem of microbial resistance. Expand the Introduction with this information, then it will be juicier.
Reply: The problem of antimicrobial resistance has been further extended in the introduction
- the name Enterobacteriaceae is no longer used, only the family is now called Entebacterales.
Reply 1: corrected
- Salmonella should be in italics.
Reply: corrected
- In writing the name: Salmonella enterica subsp. enterica does not need to use the full word "subspecies".
Reply: corrected
- When using the name Salmonella Typhimurium, the full name should be used for the first time, ie Salmonella enterica subsp. enterica serovar Typhimurium. The short name may be used only in the next use. Please, let the Authors refer to the correct naming of the bacteria.
Reply: corrected
- The signature to Fig 1 is on a different page than the Figure itself.
Reply: corrected
- No caption for Table 1. The name "Table 1" is in the Result description string.
Reply: caption for Table 1 has been added
- Table 1 is broken, the column widths should be changed.
Reply: the original format for table 1 implied for table itself a horizontal page. When submitted the horizontal page is automatically converted in vertical. The table has now been reorganised to solve the above-mentioned problem.